# Wealth and Education Inequities in Maternal and Child Health Services Utilization in Rural Ethiopia

**DOI:** 10.3390/ijerph19095421

**Published:** 2022-04-29

**Authors:** Alem Desta Wuneh, Afework Mulugeta Bezabih, Yemisrach Behailu Okwaraji, Lars Åke Persson, Araya Abrha Medhanyie

**Affiliations:** 1School of Public Health, College of Health Sciences, Mekelle University, Mekelle P.O. Box 1871, Ethiopia; afework.mulugeta@gmail.com (A.M.B.); araya.medhanyie@gmail.com (A.A.M.); 2London School of Hygiene & Tropical Medicine, London WC1E 7HT, UK; yemisrach.okwaraji@lshtm.ac.uk (Y.B.O.); lars.persson@lshtm.ac.uk (L.Å.P.); 3Ethiopian Public Health Institute, Addis Ababa P.O. Box 1242, Ethiopia

**Keywords:** inequity, antenatal care, skilled assistance at delivery, full child immunization, maternal education, household wealth, interaction

## Abstract

As part of the 2030 maternal and child health targets, Ethiopia strives for universal and equitable use of health services. We aimed to examine the association between household wealth, maternal education, and the interplay between these in utilization of maternal and child health services. Data emanating from the evaluation of the Optimizing of Health Extension Program intervention. Women in the reproductive age of 15 to 49 years and children aged 12–23 months were included in the study. We used logistic regression with marginal effects to examine the association between household wealth, women’s educational level, four or more antenatal care visits, skilled assistance at delivery, and full immunization of children. Further, we analyzed the interactions between household wealth and education on these outcomes. Household wealth was positively associated with skilled assistance at delivery and full child immunization. Women’s education had a positive association only with skilled assistance at delivery. Educated women had skilled attendance at delivery, especially in the better-off households. Our results show the importance of poverty alleviation and girls’ education for universal health coverage.

## 1. Introduction

The 2030 Sustainable Development Goals (SDGs) call for reducing maternal mortality to less than 70 per 100,000 live births, under-five mortality to not more than 25/1000 live births, and neonatal mortality to 12/1000 live births or less. The third development goal is set to ensure equitable health coverage across all low-, middle-, and high-income countries [1]. Despite the decline in maternal and under-five mortality during the Millennium Development Goals era, high mortality rates prevail in sub-Saharan Africa [2], to a large extent avoidable [3], and linked to social determinants [4]. To reach the mortality targets and the goal of universal health coverage [5], these health inequities must be addressed [6]. 

Many studies have described socioeconomic inequities in maternal and child health services utilization [7,8]. Better-off households and educated parents are more likely to use the health services, while the poor and those without education show low coverage of services [9]. Providing equitable maternal and child health care services, such as antenatal care and skilled birth assistance, improves maternal health and reduces maternal and child mortality [2]. Inequities in the coverage of these services are prominent in sub-Saharan Africa [10]. An analysis from three sub-Saharan African countries, including Ethiopia, described persistent inequities despite an increase in the coverage of essential maternal and newborn care interventions [11]. Common social determinants linked to inequities in the utilization of maternal and child health services are economic conditions, parents’ education level, the mothers’ age, and socio-cultural factors [12]. In Ethiopia, the poorest households show the lowest coverage of maternal, newborn, and child health services [13]. Women’s educational level, household wealth, and maternal age have been reported as determinants of health service utilization [14]. 

Based on a recent survey of mainly rural populations in four Ethiopian regions, we did not find evidence of wealth-based inequity in full child immunization coverage [15]. In contrast, an analysis based on national-level data found inequities related to household wealth and maternal education [16]. While inequities in maternal, newborn, and child healthcare services utilization frequently have been described, most studies report unidimensional analyses, either related to household wealth [15] or maternal education [17] but without exploring the association with the combination of household wealth and maternal education levels [14,18]. 

Hence, this study aimed to examine the association between household wealth, maternal education, and the interplay between household wealth and maternal education on the utilization of four or more antenatal visits, skilled assistance at delivery, and full child immunization. 

## 2. Methods 

### 2.1. Study Design and Setting

A cross-sectional study was conducted in 46 districts of four Ethiopian regions, i.e., Amhara, Oromia, Southern Nations, Nationalities and Peoples Region, and Tigray, from December 2018 to February 2019. These regions are the most populous in the country, where the Ethiopian Government initiated the Optimizing the Health Extension Program interventions. The survey was conducted jointly by the London School of Hygiene and Tropical Medicine, the Ethiopian Public Health Institute, and four Ethiopian universities; the University of Gondar, Jimma, Mekelle, and Hawassa Universities. The country has a three-tiered health system with primary healthcare units and secondary and tertiary levels of care. The population size of the study districts was on average 130,000 people, with 23% being women of the reproductive age and 20% children below the age of five years. One-third of the districts had a hospital. There were, on average, five health centers per district and five health posts under each health center [19].

### 2.2. Data Source

This study used data from the evaluation of the Optimizing the Health Extension Program intervention that aimed at improving services utilization. A two-stage stratified cluster sampling technique was used to select study subjects. First, 194 enumeration areas, the primary sampling unit, were obtained based on the 2007 Ethiopian Housing and Population Census using probability proportional to the size of the districts. Second, all households within the clusters were listed. Sixty households per cluster were selected using systematic random sampling. All women of reproductive age (15–49 years old) and children under the age of five years, who lived in the selected households, were included in the study. A standard sample size formula was used to calculate the sample size. The sample size was estimated to be 6000 households per group (12,000 in total). The sample size determination was detailed elsewhere [20]. 

The questionnaire was developed based on existing large-scale survey tools in English, translated into local language and back translated and pretested. Data collectors were trained for 10 days including field training before the start of data collection. Information about antenatal care attendance and delivery by skilled birth assistance was collected from all reproductive-age women who had a live birth during one year preceding the survey. Immunization information was collected by combining data recorded on children’s vaccination cards and responses from the parents if the vaccination card was missing. The questionnaire also included information on sociodemographic data and household assets. Data were collected on personal digital assistants (Companion Touch 8), and tablets (Toshiba and Hewlett Packard) programmed with CSPro 7.1. through face-to-face interviews. Data collectors sent encrypted data from the field to the password-protected server at the Ethiopian Public Health Institute. Data managers conducted quality checks and provided feedback to field teams. Data were cleaned and checked for consistency and completeness. 

### 2.3. Measurements 

#### Outcome Variables 

The analysis included three maternal and child health indicators: four or more antenatal care visits, skilled birth assistance, and full immunization of children aged 12–23 months. Four or more antenatal care visits were defined as the percentage of women of reproductive age with a live birth within the last 12 months preceding the survey who attended four or more antenatal care visits during pregnancy. Skilled birth assistance was represented in the percentage of women aged 13–49 years with a live birth within the last 12 months preceding the survey who were attended at delivery by skilled health personnel. Full immunization was defined as the percentage of children aged 12–23 months who had received one dose of BCG vaccine, three doses of polio vaccine, three doses of pentavalent vaccine, and one dose of measles vaccine [21]. All these outcomes were coded as 1 when the subjects had received the service or 0 when the subjects had not received the service. 

Covariates. The covariates included in this study were household wealth, which was created by dividing the household wealth index into three equal tertiles (Tertile 1, Tertile 2, and Tertile 3) to classify households as poor, middle, and better-off. The wealth tertile was created based on ownership of durable assets, access to utilities and infrastructure, and housing characteristics. The construction of the wealth tertile was done using principal component analysis as detailed in a previous publication [19]. Maternal education was categorized into two levels: no education (not attended formal education) and educated (primary or above). Other covariates included were maternal age in years (15–24, 25–34, and 35–49), birth order (1, 2–3, and 4 and above births), region (Amhara, Oromia, SNNPR, and Tigray), religion (Orthodox Christian, Muslim, Protestant, and others), and sex of the child. Wealth-education was also created by combining household wealth and maternal education and was categorized into six levels: tertile 1*no education, tertile 1*educated, tertile 2*no education, tertile 2*educated, tertile 3*no education, and tertile 3*educated. 

### 2.4. Data Analysis 

Descriptive analyses included frequency distributions of the determinants and covariates and outcomes of the service utilization. The utilization of services was cross-tabulated with socioeconomic and other background factors. Logistic regression was used to examine the associations between household wealth, maternal education, maternal age, birth order, region and religion and outcomes of the service utilization, and interactions between household wealth and maternal education. The results from the logistic regression analyses were presented as average marginal effects with 95% confidence intervals for the main effects and 90% confidence intervals for the interaction terms. The average marginal effects were used to estimate the discrete change for the factor’s levels from the reference. The Chi-square test was used to measure the significance of the change. Potential multicollinearity between the covariates used in the multivariate regression model was assessed using variance inflation factors. The Delta method was used for the standard errors to estimate the variation. The marginal effects were estimated using the margins command in Stata 14.1 for windows (GSW) (StataCorp LLC, College Station, TE, USA), which considered the interaction terms included in the model. Marginsplot command in Stata [22] was used to graphically display the results. During the analysis, all the commands were preceded with svy to account for clustering.

Ethical review: Ethical approval was obtained from the Ethiopian Public Health Institute (SERO-012-8-2016; Version 001), London School of Hygiene and Tropical Medicine (LSHTM Ethic Ref: 11235), and the IRB office of College of Health Sciences of Mekelle University in Ethiopia (ERC 1434/2018). Written consent and assent were also obtained from the participants.

## 3. Results

### 3.1. Participants’ Characteristics 

Data were collected from 10785 rural households. A total of 1720 women who had a live birth during the year preceding the survey and 677 children in the age interval 12–23 months were included in the analysis. The women had a mean (SD) age of 28.6 (6.04) years, and about half had no education (Table 1). Moreover, 52% of women who gave birth during the year preceding the survey had received four or more antenatal care visits, and 56% of them had got skilled assistance at delivery. Utilization of four or more antenatal care visits was slightly lower than skilled assistance at delivery. Full immunization in children aged 12–23 months was 38% (Table 1).

### 3.2. Utilization of Four or More Antenatal Care, Delivery Care and Child Immunization

Receiving four or more antenatal care visits was slightly higher among women belonging to better-off households. Skilled attendance at birth was higher among women from the better-off households and the educated ones (Table 2). Similarly, children aged 12–23 months from better-off households had higher coverage of full child immunization. Full immunization did not differ by maternal education. Skilled assistance at delivery and full child immunization increased with increasing household wealth and maternal education (Table 2). 

#### Social Determinants of Antenatal Care, Skilled Birth Assistance, and Full Child Immunization

Four or more antenatal care visits did not significantly differ by household wealth levels (*p* = 0.142). Similarly, there was no significant difference in receiving four or more antenatal care visits by levels of maternal education (*p* = 0.178). However, women from better-off households had skilled assistance at delivery more frequently than those from poor households (*p* < 0.001). Similarly, educated women had skilled assistance at delivery more often than non-educated (*p* = 0.011). Children from better-off households had higher coverage of full immunization compared to children from poor households (*p* = 0.004). There was no association between maternal education and full immunization (*p* = 0.913) (Table 3). 

Household wealth and education interaction were observed for skilled assistance at delivery at higher household wealth. In better-off households, the association between women’s education and skilled assistance at delivery was more pronounced (*p* < 0.001) (Table 3). Also, educated women in the middle household wealth had higher-skilled assistance at delivery (*p* = 0.027). However, the analyses of the interaction between household wealth and women’s education on four or more antenatal care visits and full child immunization showed overlapping estimates and were therefore inconclusive (Figure 1, Figure 2 and Figure 3 and Table 3).

## 4. Discussion

In this study in rural areas of four Ethiopian regions, we have shown that household wealth was positively associated with skilled assistance at delivery and full child immunization. Women’s education had a positive association only with skilled assistance at delivery. Educated women had higher-skilled assistance at delivery, especially for those living in better-off households. However, there was no interaction between household wealth and education on four or more antenatal visits or full child immunization. Half of the pregnant women had attended antenatal care four or more times, a bit more than half had skilled assistance at delivery, and four out of ten children were fully immunized. 

We did not find any association between household wealth and four or more antenatal care visits. This indicates that the present level of antenatal care coverage is relatively equitable in the study districts in four rural Ethiopian regions. Our findings corroborate with a community-based study from north-eastern Ethiopia [23] and Myanmar [24]. However, in a study conducted two years earlier in the same geographic area [15] as well as in the Ethiopian Demographic and Health Survey 2019 [25], there were pro-rich inequities in the utilization. Other studies in African countries have also shown social differences in the use of these services [2]. Similarly, maternal education was not associated with antennal care visits. The community-based pro-poor and pro-rural policy initiatives in Ethiopia, such as the health extension workers and women’s development groups mobilize all pregnant women to use the antenatal care services. The women’s development groups are key community actors in supporting the health extension workers by identifying pregnant women and linking them with the health extension workers and health facilities for antenatal follow-up [26]. Such policy initiatives may have contributed to the equitable utilization of antenatal care by pregnant women at different levels of socioeconomic status. 

Nonetheless, household wealth was positively associated with skilled assistance at delivery. The better-off women were more likely to get skilled assistance at delivery. In Ethiopia, skilled assistance at delivery is provided at fixed health facilities staffed with skilled health workers. This implies that services delivered at higher-level health facilities by skilled providers would likely be inequitably distributed. Such divides are frequently found in studies from low- and middle-income countries [27,28]. The reasons behind this may be associated with the costs linked to the use of these services, such as buying medicines from private pharmacies [29], transport costs [17], and other opportunity costs [30]. These findings are in line with previous studies in Ethiopia [15] and a narrative review carried out in African countries [31], which found pro-rich differentials in facility delivery. Poverty is closely linked to low-skilled assistance at birth. Empowering women and poverty alleviation efforts may enhance equity in the coverage of facility delivery. Hence, the community-based health insurance and poverty alleviation initiatives in Ethiopia should be strengthened to increase equitable utilization of maternal health services. 

Educated women were more likely to get skilled assistance at delivery. The expansion of schools in rural areas has resulted in more educated women [17]. An educated woman is more able to access and use information about community-based initiatives and the benefit of maternal health services. She is empowered to make her own decisions and to seek care for herself and her children [17,31]. This implies that Ethiopia is a long way towards the universal coverage of skilled assistance as a centrality of the SDGs that still put the poor and uneducated at a disadvantage. 

Full child immunization was more common in children belonging to better-off households. This association was also found in the 2019 Ethiopian Demographic and Health Survey report [25], other Ethiopian studies [16], and in other African countries [32]. However, a study conducted two years earlier in the same geographic area could not find social inequity in immunization coverage [15]. Immunization is provided freely but costs related to transporting especially for poor mothers, and productivity loss as taking time off work to access immunization may negatively affect equitable coverage [16]. Weaknesses in monitoring and supervision could also have reduced outreach services and access to immunization [16]. Further, the civil unrest in many parts of the country may have increased inequity in immunization coverage. Conflict as a barrier to services utilization is also evidenced in a systematic review from low-income African countries [33]. Any failure in community mobilization may also result in inequities in child immunization coverage [34]. 

The highest utilization of skilled assistance at delivery was shown among women in the better-off households who were educated. This interaction between household wealth and maternal education on skilled assistance at delivery suggests that poverty alleviation combined with enhanced education efforts, especially for girls, may result in women delivering in health facilities to an extent that is even larger than expected. This is substantiated by prior studies [35] that have found better utilization of maternal health services among educated women in better-off households. This shows that the poor and uneducated women are still behind universal coverage for skilled assistance at delivery in Ethiopia. For example, a study from Afghanistan [36] has reported better utilization of skilled assistance at delivery among women belonging to the better-off and literate households. On the other hand, a study in India showed that poor, non-educated women are less likely to utilize healthcare services [37]. This may lead to adverse birth outcomes among the poor who are uneducated women [2].

The results did not show an association between full child immunization and the interaction of household wealth and maternal education. We did not also find an association between maternal education and full child immunization. In Ethiopia, immunization is delivered at health facilities, primarily at the primary care level, and at community level through outreach and occasionally via national campaigns with blanket coverage, especially in the rural areas. The health extension workers and women development groups deliver community-based immunization services through outreach [16,26]. While the women development groups mobilize and educate mothers to vaccinate their children, the health extension workers administer the vaccination. The support of women development group to health extension programme was evidenced in a study from Ethiopia [38]. The success of community health extension program has led to equitable access to immunization for children in rural areas [26]. This suggests that strengthening community-based outreach services has the potential to ensure social equities in immunization coverage, which is documented in previous systematic reviews from low- and -middle-income countries [27,39]. In the countdown report, it was also documented that community-based interventions tended to be more equitable than facility-based interventions [28]. 

In general, our findings show that the 2030 SDGs targets related to maternal and child health are left behind in rural Ethiopia. Education, economy, and their interactions are evidenced as major social determinants of inequity in maternal health services and then maternal mortality [4]. These inequities are also evidenced to deter the achievement of universal health coverage related to SDGs [40] that requires actions. The poor who are uneducated are left behind. So, the findings entail a need for intersectoral interventions; enhancing education (i.e., SDG-4), and poverty alleviation (SDG-1) programs to promote equitable universal health coverage towards SDG targets related to maternal and child health (SDG-3.8.1) [41]. The health sector interventions alone cannot reduce inequities [42]. To reinforce the Ethiopian Health Policy statement [43], intersectoral approaches and actions are needed in the policy initiatives to increase the equitable coverage of maternal, newborn, and child health services. Intersectoral approaches have been proven to promote universal health coverage [42]. 

Our study has strengths and limitations. We have contributed to the knowledge of inequities in the utilization of maternal, and child health services in Ethiopia. The study also identified the most disadvantaged population segments in the maternal and child health services utilization by household wealth and education. The study has some limitations. Our findings are relevant for the study districts in the four most populous regions in Ethiopia but may not represent the entire region or Ethiopia as a whole. The economic status of households was measured using a household wealth index that may not reflect the actual economic strength at the time of the study. The household wealth index is an established method for economic status but does not directly show the ability to cover costs associated with the use of health services. Education was also dichotomized into no education and educated due to the small sample of women with secondary or higher levels of education. In other studies, the empowerment of education is often shown for secondary or higher education levels. The interaction effect was estimated through the combination of household wealth and maternal education. The interactions may also be explained by other factors that potentially explain inequity in the utilization of maternal, and child health services. Finally, the use of cross-sectional data for our analysis limited causal inference. 

## 5. Conclusions

The findings suggest that the combination of wealth and education plays a significant role in the utilization of maternal health services. The interaction of wealth and education has a significant effect on the utilization of skilled birth attendance. While educated women belonging to wealthier households have a higher tendency of utilizing SBA. But, the interaction of wealth and education neutralizes the utilization of ANC4+ and full child immunization. This showed that the use of SBA and full immunization of the child were positively associated with the birth outcomes. The findings underscore the importance of empowering girls and women through poverty alleviation and education as goals of health programs. This, therefore, requires, policy strategies that promote intersectoral approaches and actions aimed at ensure universal coverage in maternal and child health outcomes. Future studies involving interaction of social determinants to estimate inequity in the maternal and child health services utilization should consider large sample sizes. 

## Figures and Tables

**Figure 1 ijerph-19-05421-f001:**
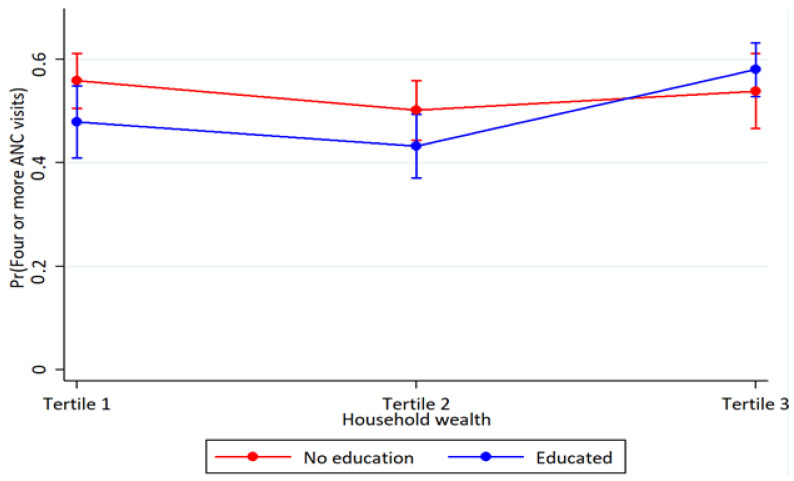
Probability of four or more antenatal care use by household wealth and maternal education with 95% confidence intervals.

**Figure 2 ijerph-19-05421-f002:**
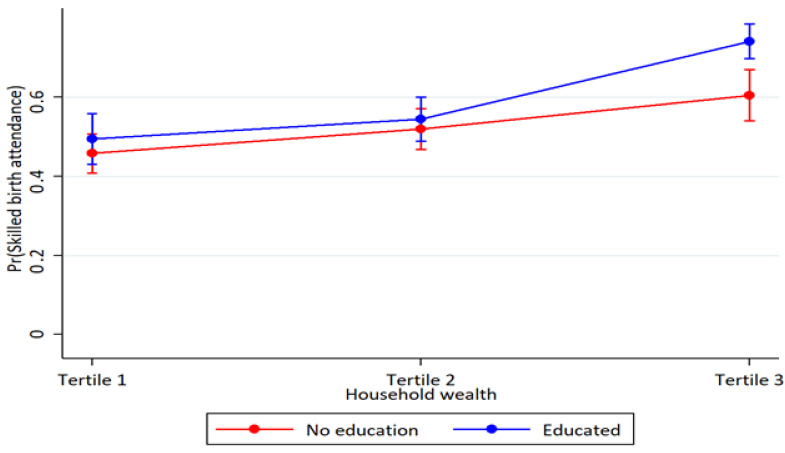
Probability of skilled birth assistance use by household wealth and maternal education with 95% confidence intervals.

**Figure 3 ijerph-19-05421-f003:**
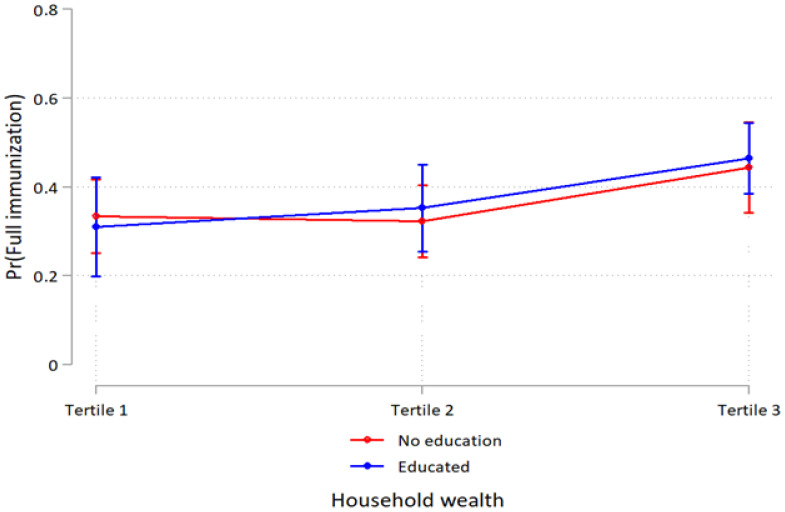
Probability of full immunization use by household wealth and maternal education with 95% confidence intervals.

**Table 1 ijerph-19-05421-t001:** Characteristics of study participants in four Ethiopian regions.

Characteristics	Frequency	%	95%CI
**Maternal characteristics (*n* = 1720)** Four or more antenatal care visits			
No	823	48	(45–50)
Yes	897	52	(50–54)
Skilled birth assistance			
No	753	44	(41–46)
Yes	967	56	(54–59)
Maternal age			
15–24	421	24	(22–26)
25–34	980	57	(55–59)
35–49	319	19	(17–20)
Household wealth			
Tertile 1 (Poor)	585	34	(32–36)
Tertile 2	571	33	(31–35)
Tertile 3 (Better-off)	564	33	(31–35)
Maternal education			
No education	872	51	(48–53)
Educated	848	49	(47–52)
Birth order			
1 birth	274	16	(14–18)
2–3 births	618	36	(34–38)
≥4 births	823	48	(46–50)
Religion			
Orthodox Christian	942	55	(52–57)
Muslim	504	29	(27–32)
Protestant and others	274	16	(14–18)
Region			
Amhara	563	33	(30–35)
Oromia	661	38	(36–41)
SNNPR *	196	11	(10–13)
Tigray	300	18	(16–19)
**Child characteristics (*n* = 677)**			
Immunization			
Not fully immunized	422	62	(59–66)
Fully immunized	255	38	(34–41)
Sex of child			
Boy	350	52	(48–55)
Girl	327	48	(45–52)

* Southern Nations, Nationalities and Peoples Region.

**Table 2 ijerph-19-05421-t002:** Utilization of maternal, newborn and child health services by household wealth, maternal education and their interactions.

Determinants	Antenatal Care Four or More Visits (*n* = 1715)		Skilled Birth Assistance (*n* = 1715)		Full Child Immunization (*n* = 677)	
	%	*p*-Value	%	*p*-Value	%	*p*-Value
**Household wealth**						
Tertile 1 (Poor)	54		40		30	
Tertile 2	47	0.005	55	0.000	34	0.000
Tertile 3 (Better-off)	56		75		47	
**Maternal education**						
No education	54	0.237	52	0.000	37	0.942
Educated	51		61		38	
**Wealth*education**						
Tertile 1*no education	57		39		32	
Tertile 1*educated	48		41		27	
Tertile 2*no education	50	0.003	57	0.000	34	0.003
Tertile 2*educated	43		52		34	
Tertile 3*no education	53		68		50	
Tertile 3*educated	58		78		46	

**Table 3 ijerph-19-05421-t003:** Probability of utilization of four or more antenatal care visits, skilled attendance at delivery, and full immunization of children aged 12–23 months by household wealth, maternal education, and their interaction. Average marginal effect estimates (main effects and interactions) based on multivariable logistic regression.

Covariates	Four or More Antenatal Care Visits	Skilled Birth Assistance	Full Immunization
	AME (95%CI)	AME (95%CI)	AME (95%CI)
**Main effects**			
Household wealth			
Tertile 1 (Poor)	Referent	Referent	Referent
Tertile 2	−0.06 (−0.12–0.003)	0.05 (−0.00–0.11)	−0.01 (−0.09–0.08)
Tertile 3 (Better-off)	0.05 (−0.02–0.11)	0.21 (0.15–0.26) ***	0.14 (0.04–0.23) **
Maternal education			
No education	Referent	Referent	Referent
Educated	−0.04 (−0.09–0.02)	0.06 (0.01–0.11) *	0.01 (−0.07–0.09)
**Joint effects (wealth*education)**			
Tertile 1*no education	Referent	Referent	Referent
Tertile 1*educated	−0.08 (−0.17–0.01)	0.04 (−0.04–0.12)	−0.02 (−0.16–0.11)
Tertile 2*no education	−0.06 (−0.13–0.02)	0.06 (−0.01–0.13)	−0.01 (−0.11–0.10)
Tertile 2*educated	−0.13 (−0.21–0.04)	0.09 (−0.01–0.16) **	0.02 (−0.11–0.15)
Tertile 3*no education	−0.02 (−0.11–0.07)	0.15 (0.06–0.23) ***	0.11 (−0.02–0.24)
Tertile 3*educated	0.02 (−0.06–0.10)	0.28 (0.21–0.35) ***	0.13 (0.01–0.25) *

AME = Average Marginal Effect = the discrete change from the base level. Logistic regression, adjusted for maternal age, birth order, region, religion, sex of a child. 95% confidence intervals in brackets, * *p* < 0.05, ** *p* < 0.01, *** *p* < 0.001. Comparison is made against the referent individuals.

## Data Availability

Deidentified data may be available from the data repository at EPHI upon request to Mrs. Martha Zeweldemariam, email: martha.zeweldemariam@lshtm.ac.uk.

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
