# Peer review of "Wealth and Education Inequities in Maternal and Child Health Services Utilization in Rural Ethiopia"

_ijerph, 2022, doi:10.3390/ijerph19095421_

Round 1

Reviewer 1 Report

This is an interesting study examining the wealth and education inequities in maternal and child health services utilization in rural Ethiopia using cross sectional study design approach. The research question seems to be important and, in general, the employed methodology is sound. My specific comments are stated below; 

Abstract:
Within the abstract, the authors need to communicate the mean age (and possibly age range) of the participants.

Methods:

Ethical approval – Please, provide information about ethical approval for this study under the method section. Which institutional review board (IRB)/ethical review committee approved the conduct of the study? Also please provide the approval number if an IRB approved the conduct of the study or survey that provided the data that this study used. Research involving human participants and/or human tissues require ethical approval by an IRB.

Please provide the method you used to deal with potential multicollinearity among independent variables.

Did the authors adjust for data structure clustering using a multi-level mixed (fixed and random effects) modelling approach considering the hierarchical nature of data from the evaluation of the Optimizing the Health Extension Program intervention used? If not, this should be considered.

Also, please provide the method used to validate your final model (k-fold cross-validation? bootstrap? etc)

It will be a good idea that the authors state if the Stata version 14.1 used was windows or Mac version.

Discussion/Conclusion: 

Can causal inference be drawn in this study since the study design is cross-sectional in nature? Please, add this to the discussion section.

Also, this reader believes the discussion section in this paper is a bit weak. An expansion on what the findings mean and the implications for maternal and child health outcomes in Ethiopia should be communicated in this section in greater detail.

Recommendation: Major revision

Author Response

The Editor

We wish to express our appreciation to the reviewers for their comments and suggestions, which have helped us significantly improve our manuscript entitled Wealth and education inequities in maternal and child health services utilization in rural Ethiopia. We appreciate your time and effort in providing valuable feedback on our manuscript. We have incorporated changes in the manuscript reflecting suggestions provided by the reviewers and highlighted these changes within the manuscript. Below is a point-by-point response to the reviewers’ comments.

Sincerely yours

For the group of authors

Alem Desta Wuneh

Responses to Comments from Reviewer 1

Comments and Suggestions for Authors

This is an interesting study examining the wealth and education inequities in maternal and child health services utilization in rural Ethiopia using cross sectional study design approach. The research question seems to be important and, in general, the employed methodology is sound. My specific comments are stated below; 

Abstract:
Within the abstract, the authors need to communicate the mean age (and possibly age range) of the participants.

Response: Thank you for your comment. We now reported the age range of the participants in the abstract.

Methods:

Ethical approval – Please, provide information about ethical approval for this study under the method section. Which institutional review board (IRB)/ethical review committee approved the conduct of the study? Also please provide the approval number if an IRB approved the conduct of the study or survey that provided the data that this study used. Research involving human participants and/or human tissues require ethical approval by an IRB.

Response: In the previous version of the manuscript, we reported this at the end of manuscript. We have now moved this to the methods section line 149-152 page 4.

Please provide the method you used to deal with potential multicollinearity among independent variables.

Response: Thank you. We have used variance inflation factor to address the multicollineairty. Following your suggestion, we have added a sentence, mentioning that potential multicollinearity between the co-variates used in the multivariable regression model was assessed using variance inflation factor. Line 140-142 page 3.

Did the authors adjust for data structure clustering using a multi-level mixed (fixed and random effects) modelling approach considering the hierarchical nature of data from the evaluation of the Optimizing the Health Extension Program intervention used? If not, this should be considered.

Response: You have raised an important point here. Yes, the multi-level model is a good model for displaying the hierarchical nature of data. However, this is beyond the scope of our study interest, which didn’t include context as a measure. Prior to the analysis, we tested the dependence using correlation matrix of the observations showing independency, i.e., the residuals were not correlated to each other. So, for these reasons, we used a single-level model for the individual-level data to estimate single-level outcome, i.e., inequity in the utilization of maternal and child health services. Our research question was what levels of care utilization you have if wealth and edcuation were combined (intesectionality of both variables). To answer this question, we ran multivariable logistic regression as a basic model combined with marginal effects as the final model. We have remained counfounding by not included factors as a limitation in the limitation paragraph in the discussion section. Line 344-346 page 11.

Also, please provide the method used to validate your final model (k-fold cross-validation? bootstrap? etc).

Response: We did not run a multi-level mixed modeling. As a basic model we ran multivariable logistic regression followed by marginal effect analysis. In this analysis, delta method was used for the standard errors to estimate the variation. This is now mantioned in the methods section, line 142-143 page 3.

It will be a good idea that the authors state if the Stata version 14.1 used was windows or Mac version.

Response: We used the Stata version 14.1 for windows (GSW). Now mentioned in line 144 and page 3.

Discussion/Conclusion: 

Can causal inference be drawn in this study since the study design is cross-sectional in nature? Please, add this to the discussion section.

Response: We agree with you; in a cross-sectional study causal inferences can not be drawn. We have mentioned this as a limitation in the discussion section line 346-347 page 11.

Also, this reader believes the discussion section in this paper is a bit weak. An expansion on what the findings mean and the implications for maternal and child health outcomes in Ethiopia should be communicated in this section in greater detail.

Response: Thank you for your constructive comment. Regarding the implication, we have addressed the overall implication of the findings in the last paragraph of the discussion (lines 318-330 page 11). For each of the main findings, we have mentioned what they imply in the context of the Sustainable Development Goals and targets related to maternal and child health. Ethiopia is lagging behind in these regards.

Recommendation: Major revision.

Response: Thank you for your constructive comments.

Reviewer 2 Report

This study aimed to determine the association between household wealth, maternal education, and the interplay between these two on the utilization of maternal and child health services. The manuscript is well structured and well written. While the results from this study are of importance as they can be used to promote the use of these services, some concerns and suggestions should be appropriately addressed before publication.

  1. Categorization of maternal education should be further elaborate, e.g., at what educational level mothers were classified as “no education”.
  2. Is it possible to subcategorize maternal education into several levels e.g., high school, undergraduate..and so on?
  3. Can you provide examples of interactions contributing to inequity in the utilization of maternal and child health services that should be further investigated.
  4. Recommendations for future studies that can minimize or alleviate the limitations of this study should also be stated.

Author Response

The Editor

We wish to express our appreciation to the reviewers for their comments and suggestions, which have helped us significantly improve our manuscript entitled Wealth and education inequities in maternal and child health services utilization in rural Ethiopia. We appreciate your time and effort in providing valuable feedback on our manuscript. We have incorporated changes in the manuscript reflecting suggestions provided by the reviewers and highlighted these changes within the manuscript. Below is a point-by-point response to the reviewers’ comments.

Sincerely yours

For the group of authors

Alem Desta Wuneh

Responses to Comments from Reviewer 2

Review Report Form

Yes

Can be improved

Must be improved

Not applicable

Does the introduction provide sufficient background and include all relevant references?

(x)

( )

( )

( )

Is the research design appropriate?

(x)

( )

( )

( )

Are the methods adequately described?

( )

(x)

( )

( )

Are the results clearly presented?

(x)

( )

( )

( )

Are the conclusions supported by the results?

( )

(x)

( )

( )

Comments and Suggestions for Authors

This study aimed to determine the association between household wealth, maternal education, and the interplay between these two on the utilization of maternal and child health services. The manuscript is well structured and well written. While the results from this study are of importance as they can be used to promote the use of these services, some concerns and suggestions should be appropriately addressed before publication.

  1. Categorization of maternal education should be further elaborate, e.g., at what educational level mothers were classified as “no education”.

Response 1: Thank you for your positive feedback on the overall manuscript. Regarding maternal edcuation, we dichotomozed data into “no education” and “educated”. The “no education” represented women who never had attended any formal education. The category “educated” included women with primary or higher level of edcuation.

  1. Is it possible to subcategorize maternal education into several levels e.g., high school, undergraduate..and so on?

Response: The number of women with secondary or higher levels of education was very small and compelled us to dichotomize maternal education into “not educated” and “educated”, as replied in the previous question. We have addressed this issue in the limitation part of the discussion section line 340-343 page 11.

  1. Can you provide examples of interactions contributing to inequity in the utilization of maternal and child health services that should be further investigated. The interaction of education and maternal age. 

Response: The small number of women in the older age groups prevented us from performing the suggested interaction analysis.

  1. Recommendations for future studies that can minimize or alleviate the limitations of this study should also be stated.

Response: Future studies involving interaction of social determinants to estimate inequity in the maternal and child health services utilization should consider large sample sizes line 359-361 page 11.

Reviewer 3 Report

By means of a cross-sectional study, this paper examines the association between household wealth, maternal education, and the interplay between household wealth and maternal education on the utilization of four or more antenatal visits, skilled assistance at delivery and full child immunization. 

While the sampling procedure is explained in detail, the data collection is not explained: What was the role of 'data collectors'? Did they act as interviewers, or were surveys completely self-administered?

Also, the development of the questionnaire is not explained: were questions pre-tested?

In the results, I was a bit surprised to read that data was collected from 10785 rural households: why rural, when in the methods section the selected regions were described as most populous in the country? In the discussion the areas are also referred to as rural rather than populous.

Analyses are described clearly. I am only wondering whether correlation between maternal age and education was taken into account in the analysis: higher educated women usually give birth at a late age than lower educated women, so the association between education and skilled birth assistance might be confounded with maternal age.

Author Response

The Editor

We wish to express our appreciation to the reviewers for their comments and suggestions, which have helped us significantly improve our manuscript entitled Wealth and education inequities in maternal and child health services utilization in rural Ethiopia. We appreciate your time and effort in providing valuable feedback on our manuscript. We have incorporated changes in the manuscript reflecting suggestions provided by the reviewers and highlighted these changes within the manuscript. Below is a point-by-point response to the reviewers’ comments.

Sincerely yours

For the group of authors

Alem Desta Wuneh

Responses to Comments from Reviewer 3

Review Report Form

Yes

Can be improved

Must be improved

Not applicable

Does the introduction provide sufficient background and include all relevant references?

(x)

( )

( )

( )

Is the research design appropriate?

(x)

( )

( )

( )

Are the methods adequately described?

( )

( )

(x)

( )

Are the results clearly presented?

( )

( )

(x)

( )

Are the conclusions supported by the results?

( )

(x)

( )

( )

Comments and Suggestions for Authors

By means of a cross-sectional study, this paper examines the association between household wealth, maternal education, and the interplay between household wealth and maternal education on the utilization of four or more antenatal visits, skilled assistance at delivery and full child immunization. 

While the sampling procedure is explained in detail, the data collection is not explained: What was the role of 'data collectors'? Did they act as interviewers, or were surveys completely self-administered?

Also, the development of the questionnaire is not explained: were questions pre-tested?

Response: Thank you very much for your comments. The data collection was carried out in a face-to-face intervews by trained data collectors. The data collectors together with their supervisors were trained for 10 days. The questionnaire was developed based on existing large-scale survey tools (DHS and MICS) in English, translated, back-translated and  pretested. We have mentioned these issues in line 89-91 page 2.

In the results, I was a bit surprised to read that data was collected from 10785 rural households: why rural, when in the methods section the selected regions were described as most populous in the country? In the discussion the areas are also referred to as rural rather than populous.

Resonse: The regions where this study was conducted were project areas for the Optimizing Health Extension Program intervention that aimed at improving health services utiization. The government initiated the project in the selected rural districts of the regions. This is mentioned in line 67-69 page 2.

Analyses are described clearly. I am only wondering whether correlation between maternal age and education was taken into account in the analysis: higher educated women usually give birth at a late age than lower educated women, so the association between education and skilled birth assistance might be confounded with maternal age.

Response: We tested the correlation between maternal age and education, showing a weak inverse correlation. We also ran an interaction analysis between maternal age and education, but the small numbers at higher education and older age prevented a proper analysis. The maternal age was included in the multivariate analysis to accounf for any confounding by age and was mentioned in line3 134-136 page 3.

Reviewer 4 Report

Knowing the characteristics of populations, economics, health education and both prenatal and vaccination assistance programs in rural areas of developing countries, it is possible to determine how the wealth of the home, the use of qualified assistance in childbirth and the complete immunization of the child was positively associated with maternal-fetal outcomes. The findings underscore the importance of empowering girls and women through poverty alleviation and education as goals of health programs.

Author Response

The Editor

We wish to express our appreciation to the reviewers for their comments and suggestions, which have helped us significantly improve our manuscript entitled Wealth and education inequities in maternal and child health services utilization in rural Ethiopia. We appreciate your time and effort in providing valuable feedback on our manuscript. We have incorporated changes in the manuscript reflecting suggestions provided by the reviewers and highlighted these changes within the manuscript. Below is a point-by-point response to the reviewers’ comments.

Sincerely yours

For the group of authors

Alem Desta Wuneh

Responses to Comments from Reviewer 4

Reviewer 4

Review Report Form

Yes

Can be improved

Must be improved

Not applicable

Does the introduction provide sufficient background and include all relevant references?

(x)

( )

( )

( )

Is the research design appropriate?

(x)

( )

( )

( )

Are the methods adequately described?

(x)

( )

( )

( )

Are the results clearly presented?

(x)

( )

( )

( )

Are the conclusions supported by the results?

(x)

( )

( )

( )

Comments and Suggestions for Authors

Knowing the characteristics of populations, economics, health education and both prenatal and vaccination assistance programs in rural areas of developing countries, it is possible to determine how the wealth of the home, the use of qualified assistance in childbirth and the complete immunization of the child was positively associated with maternal-fetal outcomes. The findings underscore the importance of empowering girls and women through poverty alleviation and education as goals of health programs.

Response: Thank you very much for a nice summary of our study findings.

Round 2

Reviewer 1 Report

Some of the issues raised in the initial review have been addressed and the manuscript is improved

Reviewer 3 Report

All comments were addressed well by the authors.